# Assessment of Factors Associated with Misperception of Body Weight and Body Weight Modifications Intentions Among Adults from Saudi Arabia: A Cross-Sectional Study

**DOI:** 10.3390/healthcare13151817

**Published:** 2025-07-25

**Authors:** Ibrahim M. Gosadi

**Affiliations:** Department of Family and Community Medicine, Faculty of Medicine, Jazan University, Jazan 45142, Saudi Arabia; gossady@hotmail.com

**Keywords:** body weight, perception, weight modification, Jazan, Saudi Arabia

## Abstract

**Background/Objectives**: Misperception of body weight might be associated with the likelihood of being in a pre-contemplation phase and with a limited intention to initiate a healthy behavioral change toward weight management. The current study investigates factors associated with the misperception of body weight, body weight satisfaction, and intentions for body weight modifications. **Methods**: This study utilized a cross-sectional design to reach adults from Jazan, in the southwest of Saudi Arabia. The data collection tool measured demographics, the participants’ latest height and weight, body weight perception, satisfaction, and intentions concerning body weight modification. Chi-squared tests were used to compare the demographic characteristics between those who had correct perception and those who had a wrong one. **Results**: A sample of 685 adult participants was included in the current analysis. The mean age of the participants was 31.3 years (standard deviation: 11.1). The proportion of female participants was 52%. Fifty-four percent of the participants had a wrong perception of their body weight. Gender, employment, smoking, khat chewing, having a diagnosed condition, and BMI levels were statistically associated with the perception of body weight (*p*-values < 0.05). **Conclusions**: The detected misestimating, especially underestimation, might be associated with the likelihood of participants being in a precontemplation phase and with a limited intention to initiate a healthy behavioral change toward weight management. The practical implications of these findings indicate the importance of incorporating the assessment of weight perception alongside actual BMI measurement in clinical settings.

## 1. Introduction

Malnutrition can be bidirectional, where excess or deficiency in food intake can lead to either being underweight or overweight and obese [1]. Perception of body weight and body image might impact satisfaction and subsequent behavior toward body weight control. On some occasions, wrong perception can lead to improper satisfaction or dissatisfaction with body weight and may result in the adoption of unhealthy practices.

Body weight abnormalities are a global health concern, especially with the high prevalence of overweight and obesity [1], and the associated rising incidence of chronic noncommunicable diseases and related mortalities. According to the World Health Organization, noncommunicable diseases were responsible for the deaths of over 43 million individuals in the year 2021, with cardiovascular diseases accounting for most of these mortalities [2]. Unhealthy lifestyle factors, including raised consumption of salt [3], carbohydrates [4,5], and fats [6], low levels of physical activity [7], and being overweight or obese [8], are major contributors to the increasing impact of noncommunicable diseases.

Promotion of the importance of healthy body weight is beneficial in increasing awareness about reducing the risk of chronic diseases, preventing premature deaths, prolonging life expectancy, and enhancing quality of life. Nonetheless, despite awareness of the benefits of promoting healthy body weight and proper lifestyle, some individuals might be at risk of developing a negative and harmful perception about their body weight. The increased use and influence of social media platforms might affect how people think about their body weight, leading to body shaming and stigmatization, and may induce problematic behavior related to weight [9]. Distorted perceptions of body weight might lead to either overestimation or underestimation.

Overestimation can lead to setting unrealistic weight loss goals and adopting unhealthy dietary habits [10]. On the contrary, some might underestimate their body weight, leading to higher satisfaction levels even when having excess weight [11]. This normalization of excess body weight can result in the adoption of unhealthy lifestyles and reduce the willingness to adopt weight control measures.

Willingness to change a behavior in order to adopt a healthy life can be explained by the transtheoretical model. This model is composed of six stages which are precontemplation, contemplation, preparation, action, maintenance, and termination [12]. The normalization of abnormal body weight and the associated misperception can leave some individuals in the precontemplation stage and lower their probability of adopting healthy behavior. Therefore, this might reduce the awareness about the importance of the required modification of the behavior and may delay the progress toward the remaining stages of a change in lifestyle.

According to the American College of Lifestyle Medicine, a healthy lifestyle involves six main elements: nutrition, physical activity, stress management, high-quality sleep, social connection, and avoidance of risky substances [13]. It can be postulated that distorted body perception may have a direct correlation with the majority of these elements. For example, weight loss management is believed to involve the cognitive characteristics of behavioral change [14,15].

Exposure to risky substances which can affect lifestyle varies according to populations. One of the studied substances that can impact lifestyle is Khat chewing which is mainly practiced in eastern countries in Africa [16] and southwestern regions of the Arabian peninsula [17,18]. Additionally, Khat chewing is also practiced among migrants living in Australia [19], USA [20], and European countries [21,22,23]. Khat has a stimulant characteristic as it contains cathinone. Additionally, Khat chewing is linked to several health consequences including mental, oral and gastric, cancer, metabolic, and nutritional conditions [24,25].

The observed influence of body weight perception on how individuals evaluate their weight status can be suggested as a primary screening measure for managing body weight abnormalities. This step can assist in identifying the stage of behavior change and assessing readiness to maintain healthy practices toward weight management. The current study measures how people view their body weight in relation to their estimated body mass index (BMI). Additionally, the study investigates the concordance between actual and perceived BMI and assesses factors associated with a wrong perception of body weight.

## 2. Materials and Methods

### 2.1. Study Design and Context

This assessment is part of an investigation that targeted adults in the Jazan region to assess BMI profile, perceptions, and methods of modifying body weight. This study utilized a cross-sectional design to reach adults from Jazan, in the southwest of Saudi Arabia. Adults living in the Jazan region were included regardless of nationality or gender. Exclusion criteria were limited to being under 18 or refusing to provide consent to participate. The measurement was conducted between November 2023 and May 2024. Participation was anonymous and voluntary, and data were collected after securing ethical approval from the Standing Committee for Scientific Research of Jazan University (number REC-44/06/446, dated December 2022).

### 2.2. Data Collection Tool

The developed data collection tool had five main components. The first component measured the main demographics of the participants, including age, gender, area of residence, education level, employment, monthly income, smoking, khat chewing, and having a diagnosed medical condition. The second component involved asking about the participants’ latest height and weight. The third component asked how the participants perceived their body weight—as low body weight, normal body weight, overweight, or obese. The fourth item asked about the degree of satisfaction with their current body weight. Finally, the participants were asked about their intentions concerning body weight—either to modify their body weight or to have no intention of modification.

### 2.3. Data Collection Process

The questionnaire was converted into an online format using Google Forms. A web link was generated and distributed on social media platforms to reach the required sample. The first page of the form contained an information sheet describing the study’s purpose and the steps to allow individuals to provide informed consent. Those who consented were granted access to the questionnaire, and those who refused were directed elsewhere.

Sampling for this investigation was not random and was mainly based on convenience sampling. Participants who agreed to take part were encouraged to share the web link with other potential participants. The sample size estimation was based on findings from a similar study conducted in Saudi Arabia, which found that 66% of adolescents had a wrong perception of their body weight [26]. The StatCalc function of Epi Info was used to estimate the sample size. A sample of 595 was calculated, assuming a 66% prevalence of adults with a wrong perception of their body weight, a 99% confidence level, and a 5% margin of error.

### 2.4. Data Analysis

Data analysis was conducted using the Statistical Package for Social Sciences, version 25. Frequencies and proportions were used to summarize binary and categorical variables. Mean and standard deviation (SD) were used to summarize continuous variables. BMI was estimated by dividing the reported weight in kilograms by the reported squared height in meters. The estimated BMI values were categorized into underweight, normal weight, overweight, and obese according to World Health Organization classification guidelines [27]. Cross-tabulation was used to assess concordance between body weight perception, as reported by the participants, and the estimated BMI category. The kappa test was used to measure the magnitude of agreement between the two measures. Furthermore, chi-squared tests were used to compare demographic characteristics between those with correct perception and those with incorrect perception. All the measured demographic variables were either binary or categorical except for age which was dichotomized based on the estimated sample mean as either 31 or younger or older than 31. A follow-up comparison was made to assess satisfaction levels according to whether participants underestimated, correctly estimated, or overestimated their body weight. Finally, a comparison was made to examine the intention to modify body weight according to BMI level. A *p*-value of 0.05 or less was considered statistically significant for the applied statistical tests.

## 3. Results

A total of 908 individuals were approached in the current investigation, of whom 223 were excluded due to refusal to participate, being under 18, or not providing the information required to estimate BMI. Ultimately, a sample of 685 adult participants was involved in the current analysis, which exceeded the estimated sample size due to the utilized online approaching method. The demographic data of the sample are displayed in Table 1. The mean age of the participants was 31.3 (SD: 11.1). The proportion of female participants was 52%, and more than half of the sample (55%) were from urban areas. Majority of the participants had a university education (77%), and 44% were employed or business owners. When participants were asked about their monthly income, the most frequent response was less than 5000 SAR (45.7%). Majority of the participants were never smokers and had never chewed khat. Less than half of the participants (43%) reported being diagnosed with a chronic condition. The estimated BMI of the participants indicated that 38% had normal body weight, 11% were underweight, and the remaining 50% were either overweight or obese.

Fifty-four percent of the participants had a wrong perception of their body weight. Among those with incorrect perceptions, 244 participants (35.6%) underestimated their body weight, whereas 130 participants (19%) overestimated it. Table 2 compares the demographics of the sample according to whether participants had a wrong or correct perception of their body weight. Gender, employment, smoking, khat chewing, having a diagnosed condition, and BMI levels were statistically associated with wrong body weight perception (*p*-values < 0.05). The frequency of women reporting wrong perceptions of body weight was higher compared to men. The frequency of incorrect perception was the lowest among students in comparison to other employment groups. Wrong perception of body weight was less common among current or former smokers and khat chewers. This may suggest that those exposed to smoking or khat chewing, either currently or previously, might be more likely to be aware of their body weight in comparison to those who were never smokers or never khat chewers. Wrong perception was also more frequent among those with a diagnosed condition. The highest frequency of wrong perception was detected among obese and underweight individuals in comparison to other BMI categories. Other demographic factors such as age, education level, area of residence, and monthly income were not associated with the misperception of body weight in the current sample.

As shown in Table 3, participants were categorized according to their estimated BMI category and cross-tabulated with their perception of body weight. It can be noted that the proportion of participants who were able to correctly identify their BMI category was 37% for those who were underweight, 56% for those with normal weight, 36.7% for those who were overweight, and 68.8% for those with obesity. The identified kappa value of agreement between perceived body weight and actual estimated BMI was 0.22, suggesting low agreement. Eighteen individuals described themselves as having low body weight, although they were categorized as overweight or obese. Among those who described themselves as having normal weight, 10% were underweight, and 33% were either overweight or obese. Underestimation of body weight was observed among the sample, where among 191 participants who were categorized as overweight, 74 participants (39%) perceived their weight as normal or low. Finally, among 152 participants who are classified as obese, 130 participants (86%) perceived themselves within lower-BMI categories, indicating a strong magnitude of underestimation.

Table 4 illustrates satisfaction with body weight according to the accuracy of perception. Ninety-one participants reported being satisfied with their weight despite underestimating their BMI. Eighty-one of those who were not satisfied with their body weight tended to overestimate it.

Table 5 and Figure 1 display the declared intention of participants concerning their body weight according to their BMI level. It can be observed that the majority of participants classified as overweight or obese intended to lose weight. However, 105 participants in these categories declared no intention to lose weight. Misconception was also observed among participants classified as normal weight or underweight, with 107 individuals indicating that they needed to lose weight—suggesting the presence of a distorted body image despite having either normal weight or being underweight.

## 4. Discussion

The current study was a cross-sectional investigation that aimed to assess how adults perceive their body weight, identify factors associated with wrong perceptions, and explore participants’ intentions regarding their body weight according to their level of BMI. More than half of the participants had a wrong perception of their body weight, with the majority tending to underestimate it. Female gender, being employed, practicing smoking or khat chewing, and being either underweight or obese were associated with higher frequencies of wrong perceptions of body weight. Underestimation of body weight appears to be associated with higher satisfaction levels with BMI despite having abnormal body weight. A misconception toward body weight is likely present and appears to influence participants’ intentions, with some of those who are overweight or obese reporting no intention to lose weight. In contrast, some participants with normal weight or underweight declared their intentions to lose weight, which might indicate the presence of a distorted body image.

The findings of the current study can be compared to similar local or international investigations. In a nationwide study conducted by Althumiri et al., which recruited a sample of 4709 participants from all regions of Saudi Arabia, it was reported that 42% of the sample misclassified their weight, with misclassification being higher among participants with abnormal body weight [28], similar to our findings. Additionally, Althumiri et al. concluded that misperception was associated with obesity, age group, educational level, and having a diagnosed chronic condition, which aligns with our findings—except for the impact of age and education, which were not significant in the current study. Nonetheless, Althumiri et al. did not detect a statistically significant association between gender and body weight misperception, which was identified as an important variable in the current study. Furthermore, in the current assessment, those who were retired or unemployed more frequently reported a wrong perception of body weight in comparison to employed or students. However, no association between weight perception and employment was detected in a similar Turkish study [29].

In a cross-sectional study that recruited a sample of 385 adolescents from Jazan, Saudi Arabia, who attempted to modify their body weight, it was reported that despite 55% of the adolescents having normal body weight, the majority of the sample were not satisfied with their weight (63%) [16]. This is similar to our findings, where almost 60% of the overall sample were not satisfied with their body weight across all BMI categories.

A low agreement between perceived body weight and actual BMI category was detected in the current study. This contrasts with the findings of a study conducted in Riyadh, Saudi Arabia, by Alhussaini et al. among female university students. Alhussaini et al. indicated that among the recruited female sample, the majority were between 18 and 29, and a kappa value of 0.635 was identified [30], compared to 0.22 detected in the current investigation. Nonetheless, the lower kappa value in the present study can be attributed to the broader demographic variability of the sample, which included different age groups, both men and women and various employment categories.

The frequency of body weight misperception was higher among females in the current study in comparison to males. This is similar to the findings of other studies which assessed perception of body weight according to gender. For example, in a Saudi study which involved a sample of 334 residents in Saudi Arabia, it was concluded that the prevalence of body weight misperception was higher among female [31], which is similar to the findings of the current study. Additionally, in another Nigerian study which involved a sample of 567 adults, it was also reported that body weight misperception was higher among females in comparison to males [32]. Similarly, studies that assessed body weight satisfaction and perception among younger populations suggested that females are more likely to be less satisfied by their body weight [33], and to overestimate their body weight [34].

The findings of the current study emphasize the importance of assessing the perception of body weight in addition to measuring weight and height and estimating BMI. This notion is supported by the findings of a large-scale Canadian study involving around 200,000 participants, where it was noted that being overweight or underweight was associated with lower life satisfaction, regardless of actual body weight [35]. The high prevalence of incorrect body weight perception detected in the current investigation was linked to improperly declared intentions concerning weight management among some participants. Additionally, some individuals reported being satisfied with their body weight despite having an abnormal weight.

In the current sample, underestimation of body weight was highly prevalent among overweight and obese individuals. This aligns with the findings of a Spanish study involving 1081 adult participants, where overweight individuals were more likely to underestimate their body weight [36]. Furthermore, a similar U.S. study that involved 744 adults and assessed the concordance between perceived body weight and actual BMI reported that, among participants with an overweight BMI, 32% perceived themselves as either normal or underweight, while among obese participants, 44% perceived themselves as either normal or overweight—indicating a non-negligible presence of underestimation [37].

In addition to the identified body weight underestimation among overweight and obese individuals, other demographic factors were associated with misperception. The minority of the current sample were either current or previous smokers (18%). Additionally, only 15% were either current or previous khat chewers. Nonetheless, the frequency of wrong perception of body weight was lower among those who were current or previous smokers or khat chewers in comparison to those who were never smokers or never khat chewers. These findings are different from the findings of another study which involved a sample of 53,447 Brazilian adolescents which concluded that the prevalence of body weight misperception (either underestimation or overestimation) was higher among adolescents who tried smoking [38]. Nonetheless, another Turkish study which involved a sample of 250 adults did not detect an impact of smoking on weight misperception [29].

It is possible to argue that the higher frequency of correct perception of body weight among smokers or previous smokers may be related to the association between smoking and body weight. The current evidence suggests that some individuals who are not satisfied with their body weight or perceive themselves as obese might tend to initiate smoking [39]. Additionally, smoking initiation is reported as a weight loss measure among some individuals [40,41]. Additionally, some who quit smoking might be at risk of weight gain after smoking cessation [42]. Studies that assessed body weight misperception according to khat chewing are lacking. However, khat chewing is a well-known risk factor for lower BMI among khat chewers [43,44]. Similarly, Khat is known for its impact on appetite suppression [45], and a higher risk of under nutrition [44].

The current study has multiple strengths and limitations. One strength lies in its ability to reach a sample with diverse demographic characteristics and correlate them with the likelihood of body weight misperception. This can inform future targeted interventions addressing abnormal body weight in the region and in similar local and international contexts. The main limitation relates to the inherent nature of self-reported weight and height which can be affected by higher risk of reporting bias in comparison to an objective clinical measurement. Another limitation is due to the nature of data collection process which was based on accessing the assessment tool in an online setting. This may reduce the ability of illiterate and older subjects to participate in the study which may have a potential impact on the generalizability of the study’s findings.

## 5. Conclusions

More than half of the participants had a wrong perception of their body weight, with the majority tending to underestimate it. Female gender, being employed, practicing smoking or khat chewing, and being either underweight or obese were associated with higher frequencies of incorrect body weight perception. The detected misperception—especially underestimation—might be linked to the likelihood of participants being in a precontemplation phase, resulting in limited intention to initiate healthy behavioral change toward weight management. The practical implications of these findings highlight the importance of incorporating weight perception assessment alongside actual BMI measurement in clinical settings. Early identification of such misconceptions and the application of appropriate cognitive therapy may be required to address body weight misperception.

## Figures and Tables

**Figure 1 healthcare-13-01817-f001:**
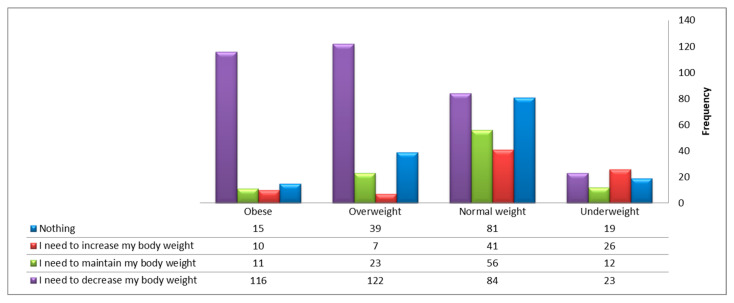
Body mass index and declared intentions for body weight modification among 685 adult participants from Jazan, Saudi Arabia.

**Table 1 healthcare-13-01817-t001:** Demographic data of 685 adult participants from Jazan, Saudi Arabia.

Variables	Frequency [Proportion]
Age *	
31 or younger	384 [56%]
Older than 31	300 [44%]
Gender	
Male	327 [47.7%]
Female	358 [52.3%]
Area of residence	
Rural	305 [44.5%]
Urban	380 [55.5%]
Education level	
School education	156 [22.7%]
Undergraduate university	487 [71.2%]
Postgraduate university	42 [6.1%]
Employment *	
Employed or business owner	304 [44.3%]
Retired or unemployed	107 [15.6%]
Student	271 [39.1%]
Monthly income	
Less than 5000 SAR	313 [45.7%]
Between 5000 and 10,000 SAR	107 [15.6%]
More than 10,000 and less than 150,000 SAR	144 [21.2%]
More than 150,000 SAR	120 [17.5%]
Smoking *	
Currently smoking	93 [13.7%]
Previously smoking	32 [4.7%]
Secondhand smoking	143 [21.1%]
Never smoker	407 [60.3%]
Khat chewing *	
Current	31 [4.5%]
Previous	72 [10.5%]
Never	581 [85%]
Diagnosed condition *	
No [Healthy]	385 [56.8%]
Yes [Diagnosed]	293 [43.2%]
Current BMI category *	
Underweight	80 [11.6%]
Normal weight	262 [38.3%]
Overweight	191 [27.9%]
Obese	152 [22.2%]

* 1 missing for age; 10 missing for smoking; 1 missing for khat chewing; 7 missing for diagnosed condition. SAR: Saudi Arabian Riyals.

**Table 2 healthcare-13-01817-t002:** Distribution of 685 adult participants from Jazan, Saudi Arabia, according to the perception of their body weight based on estimated body mass index.

Variables	Body Weight Perception Frequency [Proportion]			
	Correct	Wrong	Total	*p* Value *
Age				0.146
31 or younger	184 [47.9%]	200 [52.1%]	384 [100%]	
Older than 31	127 [42.3%]	173 [57.7%]	300 [100%]	
Gender				<0.001
Male	186 [56.9%]	141 [43.1%]	327 [100%]	
Female	125 [34.9%]	233 [65.1%]	358 [100%]	
Area of residence				0.398
Rural	133 [43.6%]	172 [56.4%]	305 [100%]	
Urban	178 [46.8%]	202 [53.2%]	380 [100%]	
Education level				0.820
School education	71 [45.5%]	85 [54.5%]	156 [100%]	
Undergraduate university	219 [45%]	268 [55%]	487 [100%]	
Postgraduate university	21 [50%]	21 [50%]	42 [100%]	
Employment				0.027
Employed or business owner	143 [47%]	161 [53%]	304 [100%]	
Retired or unemployed	36 [33.6%]	71 [66.4%]	107 [100%]	
Student	131 [48.3%]	140 [51.7%]	271 [100%]	
Monthly income				0.931
Less than 5000 SAR	145 [46.3%]	168 [53.7%]	313 [100%]	
Between 5000 and 10,000 SAR	50 [46.7%]	57 [53.3%]	107 [100%]	
More than 10,000 and less than 150,000 SAR	64 [44.4%]	80 [55.6%]	144 [100%]	
More than 150,000 SAR	52 [43.3%]	68 [56.7%]	120 [100%]	
Smoking				<0.001
Currently smoking	49 [52.7%]	44 [47.3%]	93 [100%]	
Previously smoking	23 [54.8%]	19 [45.2%]	42 [100%]	
Secondhand smoking	81 [56.6%]	62 [43.4%]	143 [100%]	
Never smoker	158 [38.8%]	249 [61.2%]	407 [100%]	
Khat chewing				0.015
Current	19 [61.3%]	12 [38.7%]	31 [100%]	
Previous	41 [56.9%]	31 [43.1%]	72 [100%]	
Never	250 [43%]	331 [57%]	581 [100%]	
Diagnosed condition				0.005
No [Healthy]	193 [50.1%]	192 [49.9%]	385 [100%]	
Yes [Diagnosed]	115 [39.2%]	178 [60.8%]	293 [100%]	
Current BMI category				<0.001
Underweight	34 [42.5%]	46 [57.5%]	80 [100%]	
Normal weight	143 [54.6%]	119 [45.4%]	262 [100%]	
Overweight	112 [58.6%]	79 [41.4%]	191 [100%]	
Obese	22 [14.5%]	130 [85%]	152 [100%]	

* Chi-squared tests.

**Table 3 healthcare-13-01817-t003:** Distribution of 685 adult participants from Jazan, Saudi Arabia, by body mass index and perception of their body weight.

Participants’ Perception About Their Body Weight	Estimated BMI Category *				Total
	Underweight	Normal Weight	Overweight	Obese	
I have a low body weight	34 [37.0%]	40 [43.5%]	7 [7.5%]	11 [12.0%]	92 [100%]
I have a normal body weight	26 [10.2%]	143 [55.9%]	67 [26.2%]	20 [7.7%]	256 [100%]
I am overweight	18 [5.9%]	76 [24.9%]	112 [36.7%]	99 [32.5%]	305 [100%]
I am obese	2 [6.3%]	3 [9.3%]	5 [15.6%]	22 [68.8%]	32 [100%]

* Kappa value of agreement between perceived body weight and actual estimated BMI is 0.22.

**Table 4 healthcare-13-01817-t004:** Body weight satisfaction among 685 adult participants from Jazan, Saudi Arabia.

Body Weight Satisfaction	Weight Estimation			
	Underestimate	Correct Estimation	Overestimate	Total
Strongly satisfied	39 [37.2%]	58 [55.2%]	8 [7.6%]	105 [100%]
Satisfied	52 [31.7%]	88 [53.7%]	24 [14.6%]	164 [100%]
Not sure	27 [35.1%]	33 [42.9%]	17 [22%]	77 [100%]
Unsatisfied	86 [38.7%]	82 [37%]	54 [24.3%]	222 [100%]
Strongly unsatisfied	40 [34.2%]	50 [42.7%]	27 [23.1%]	117 [100%]

**Table 5 healthcare-13-01817-t005:** Body mass index and intentions for body weight modification among 685 adult participants from Jazan, Saudi Arabia.

Intentions of the Participants Concerning Their Body Weight	BMI Category				Total
	Underweight	Normal Weight	Overweight	Obese	
Nothing	19 [12.3%]	81 [52.7%]	39 [25.3%]	15 [9.7%]	154 [100%]
I need to increase my body weight	26 [31.0%]	41 [48.8%]	7 [8.3%]	10 [11.9%]	84 [100%]
I need to maintain my body weight	12 [11.8%]	56 [54.9%]	23 [22.5%]	11 [10.8%]	102 [100%]
I need to decrease my body weight	23 [6.7%]	84 [24.3%]	122 [35.4%]	116 [33.6%]	345 [100%]
Total	80 [11.7%]	262 [38.2%]	191 [27.9%]	152 [22.2%]	685 [100%]

## Data Availability

The data presented in this study are available on request from the corresponding author.

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
