# Peer review of "Assessment of Factors Associated with Misperception of Body Weight and Body Weight Modifications Intentions Among Adults from Saudi Arabia: A Cross-Sectional Study"

_healthcare, 2025, doi:10.3390/healthcare13151817_

Round 1
Reviewer 1 Report
Comments and Suggestions for Authors
I thank the authors for an interesting article. I do have some concerns/queries.
Lines 31- 35: Are there any citable references for these statements? There are quite a few articles published on the subject.
Line 58: A definition of the theoretical construct of precontemplation would be useful to set the stage for the reader, it is a posited as a statement but how it is used in this setting and the Model behind it should be cited for context
Line 70 :This is useful in giving a definition of the theoretical construct of behaviour change of which pre-contemplation is the first phase . But this comes after the concept has been stated several times in the manuscript.
Line 90:Were objective measures of height and weight done? If not could self-reported measures also be a limitation considering the papers is estimating perception.
Line 126: Why 908 individuals? I recognize that snowball sampling was used but the calculated sample size was 595 was a high refusal rate anticipated? what was the stopping criteria?
Line 131: Is it that Jazon has a relatively young population or did the mode of data collection exclude the older population?
Line 177: why use numbers and not percentages?
Table 5 :I realize these are row percentages , would a more useful question be to find out of those within BMI categories what were the intentions of each category? ( column percentages) Confirming that this is the data displayed in figure 1?
Line 213:This study has a relatively young population , upper range seems to be mid 40s, could this be a limitation of the study?
Line 225: The study by Alhussaini et al of university females had a similar age group could the authors posit why the results were not consistent?
Line 256 : There is no discussion on the covariates , smoking was said to be related to body weight misconception but it is not discussed. Persons sometimes use smoking as a weight loss tool , could that behaviour be a aspect to consider in this population?
Comments on the Quality of English Language
English and grammar are appropriate
Author Response
Comment: I thank the authors for an interesting article. I do have some concerns/queries.
Response: The author of the manuscript appreciates the supportive comment of the reviewer. As per the reviewers’ request, we have applied multiple modifications to the revised manuscript to enhance the writing quality. All modifications are highlighted with yellow to allow easier assessment of applied changes.
Comment: Lines 31- 35: Are there any citable references for these statements? There are quite a few articles published on the subject.
Response: The author of the manuscript appreciates the comment of the reviewer. The statement was cited from reference number 1 which is referenced as the following:
- World Health Organization. Malnutrition 2024 [Available from: https://www.who.int/news-room/fact-sheets/detail/malnutrition. Accessed 19th of June 2025.
The inserted citation in the introduction is highlighted with yellow in the revised manuscript.
Comment: Line 58: A definition of the theoretical construct of precontemplation would be useful to set the stage for the reader, it is a posited as a statement but how it is used in this setting and the Model behind it should be cited for context
Response: The author of the manuscript agrees with the comment of the review. More description of the transtheoretical model is now added to the introduction while citing a reference to support the definition. Additionally, the model was applied to this relevant settings to indicate how the misperception about body weight can affect the progress toward other stages of the model. The following was added to the introduction of the revised manuscript and highlighted with yellow as the following:
‘Willingness to change a behavior in order to adopt a healthy life can be explained by the transtheoretical model. This model is composed of six stages, which are, precontemplation, contemplation, preparation, action, maintenance and termination [12]. The normalization of abnormal body weight and the associated misperception can leave some individuals in the precontemplation stage and lower their probability of adopting healthy behavior. Therefore, this might reduce the awareness about the importance of the required modification of the behavior and may delay the progress toward the remaining stages of a change of lifestyle.’
Comment: Line 70 :This is useful in giving a definition of the theoretical construct of behaviour change of which pre-contemplation is the first phase . But this comes after the concept has been stated several times in the manuscript.
Response: The author of the manuscript appreciates the comment of the review. More description of the transtheoretical model is now added to the introduction as explained in the response to the previous comment.
Comment: Line 90:Were objective measures of height and weight done? If not could self-reported measures also be a limitation considering the papers is estimating perception.
Response: The author of the manuscript agrees with the comment of the reviewer that self-reported measures are subjective and can be exposed to the risk of reporting bias. This has been indicated as a limitation in the last paragraph of the discussion and highlighted with yellow as the following:
‘The main limitation relates to the inherent nature of self-reported weight and height which can be affected by higher risk of reporting bias in comparison to an objective clinical measurement’
Comment: Line 126: Why 908 individuals? I recognize that snowball sampling was used but the calculated sample size was 595 was a high refusal rate anticipated? what was the stopping criteria?
Response: Due to the nature of data collection method, which was based on approaching the participants in online, electronic settings via sharing a generated web-link with the targeted population, the stopping criteria was mainly based on observing the actual responses of the approached participants until meeting the required sample to perform the analysis. This was performed during the assessment period while observing responses refusing the participation, or not meeting the inclusion criteria, or not providing information needed to perform the required analysis. This was explained in the first section of the results. Due to the nature of data collection method, the actual recruitment slightly exceeded the estimated sample size. The modified section of the results is heighted with yellow in the revised manuscript as the following:
‘A total of 908 individuals were approached in the current investigation, of whom 223 were excluded due to refusal to participate, being under 18, or not providing the information required to estimate BMI. Ultimately, a sample of 685 adult participants was involved in the current analysis, which exceeded the estimated sample size due to the utilized online approaching method.’
Comment: Line 131: Is it that Jazon has a relatively young population or did the mode of data collection exclude the older population?
Response: The recruitment did not apply any exclusion criterion based on the age the adult population. The mean age of the participants was 31 years (SD:11 years) which indicates reasonable representation of age groups. Nonetheless, due to the nature of the data collection, which was based on the generated web-link in order to access the study’s questionnaire, older or illiterate individuals are less likely to participate in the current analysis. This is now indicated as a limitation of the current assessment and added to the last paragraph of the discussion and highlighted with yellow in the revised manuscript as the following:
‘Another limitation is due to the nature of data collection process which was based on accessing the assessment tool in an online settings. This may reduce the ability of illiterate and older subject to participate in the study which may have a potential impact on the generalizability of the study’s findings.’
Comment: Line 177: why use numbers and not percentages?
Response: As per the request of the reviewer, percentages are now added to the indicated section of the results. The following modification was applied and highlighted with yellow in the revised manuscript:
‘Underestimation of body weight was observed among the sample, where among 191 participants who were categorized as overweight, 74 participants (39%) perceived their weight as normal or low. Finally, among 152 participants whom are classified as obese, 130 participants (86%) perceived themselves within lower-BMI categories, indicating a strong magnitude of underestimation.’
Comment: Table 5 :I realize these are row percentages , would a more useful question be to find out of those within BMI categories what were the intentions of each category? ( column percentages) Confirming that this is the data displayed in figure 1?
Response: The author of the manuscript appreciates the comment of the reviewer. It is agreed that adding proportions within BMI categories to table 5 would be useful. However, this would increase the complexity of the table and impact its readability. Therefore, figure 1 was added to display the same data provided in table 5 to allow easier and direct visual observation of intentions of body weight modification based on the BMI category. It is hereby confirmed that data presented in table 5 and figure one are the same and these methods of illustration and visualization are performed to allow clearer presentation of the findings. No modification to the revised manuscript was applied in response to this particular comment.
Comment: Line 213:This study has a relatively young population , upper range seems to be mid 40s, could this be a limitation of the study?
Response: As explained in a response to a previous comment, the recruitment did not apply any exclusion criterion based on the age among the adult population. Nonetheless, due to the nature of the data collection, which was based on the generated web-link in order to access the study’s questionnaire, older or illiterate individuals are less likely to participate in the current analysis. This is now indicated as a limitation of the current assessment and added to the last paragraph of the discussion and highlighted with yellow in the revised manuscript as the following:
‘Another limitation is due to the nature of data collection process which was based on accessing the assessment tool in an online settings. This may reduce the ability of illiterate and older subject to participate in the study which may have a potential impact on the generalizability of the study’s findings.’
Comment: Line 225: The study by Alhussaini et al of university females had a similar age group could the authors posit why the results were not consistent?
Response: We agree with the comment that Alhussaini et al targeted a sample of university female participants aged between 18 and 50. However, the sample by Alhussaini was mainly composed of younger population where 90% were younger than 30. This is different than the current study which recruited a more diverse sample including older age groups in comparison, and both male and female participants. This difference in the demographics is illustrated in the discussion section of the revised manuscript and highlighted with yellow as the following:
‘A low agreement between perceived body weight and actual BMI category was detected in the current study. This contrasts with the findings of a study conducted in Riyadh, Saudi Arabia, by Alhussaini et al. among female university students. Alhussaini et al indicated that among the recruited female sample, the majority were between 18 and 29, and a kappa value of 0.635 was identified [30], compared to 0.22 detected in the current investigation. Nonetheless, the lower kappa value in the present study can be attributed to the broader demographic variability of the sample, which included different age groups, both men and women and various employment categories.’
Comment: Line 256 : There is no discussion on the covariates , smoking was said to be related to body weight misconception but it is not discussed. Persons sometimes use smoking as a weight loss tool , could that behaviour be a aspect to consider in this population?
Response: The author of the manuscript agrees with the comment of the reviewer. The minority of the current sample were either current or previous smokers (18%). Additionally, only 15% were either current or previous khat chewers. Nonetheless, the frequency of wrong perception of body weight was lower among the current or previous smokers or khat chewers in comparison to never smokers or never khat chewers. It is also agreed that some individuals might initiate smoking as a weight loss measure. To enhance the discussion about how smoking and khat chewing is relevant to body weight perception, the following is added to the discussion of the revised manuscript and highlighted with yellow:
‘In addition to the identified body weight underestimation among overweight and obese individuals, other demographic factors were associated with the misperception. The minority of the current sample were either current or previous smokers (18%). Additionally, only 15% were either current or previous khat chewers. Nonetheless, the frequency of wrong perception of body weight was lower among the current or previous smokers or khat chewers in comparison to never smokers or never khat chewers. These findings are different from the findings of another study which involved a sample of 53,447 Brazilian adolescents which concluded that the prevalence of bodyweight misperception (either underestimation or overestimation) was higher among adolescents who tried smoking [38]. Nonetheless, another Turkish study which involved a sample of 250 adults did not detect an impact of smoking on weight misperception [29].
It is possible to argue that the higher frequency of correct perception of body weight among smokers or previous smokers may be related to the association between smoking and body weight. The current evidence suggests that some individuals who are not satisfied with their body weight or perceive themselves as obese might tend to initiate smoking [39]. Additionally, smoking initiation is reported as a weight loss measure among some individuals [40, 41]. Additionally, some who quit smoking might be at risk of weight gain after smoking cessation [42]. Studies that assessed body weight misperception according to khat chewing are lacking. However, khat chewing is a well-known risk factor for lower BMI among khat chewers [43, 44]. Similarly, Khat is known for its impact on appetite suppression [45], and the higher risk of under nutrition [44].’
The author also confirms that other covariates, such as age, gender, diagnosis with a chronic condition, employment, education, are also discussed throughout the discussion section. These sections are highlighted with yellow for easier assessment of discussions performed.
Comments on the Quality of English Language:
English and grammar are appropriate
The English could be improved to more clearly express the research.
Response: We agree with the comment of the reviewer. Author Services of MDPI will be contacted after completing the revision to enhance the writing quality of the revised manuscript.
Reviewer 2 Report
Comments and Suggestions for Authors
Several ideas are reiterated throughout the text. For instance, the link between misperception of body weight and unhealthy behaviours is mentioned multiple times (lines 34, 50, 56, and 58), which can cause redundancy rather than reinforcing the argument and leading to duplication and fragmented narrative structure.
It is interesting to understand why 32 was the age you chose to divide the study sample. A brief explanation is needed.
Is there any regional cultural reasoning for explaining women's misperception of overweight as being a normal silhouette?
The Khat chewing was a completely new concept and needed to read a supplementary reference: Alzahrani MA, Alsahli MA, Alarifi FF, Hakami BO, Alkeraithe FW, Alhuqbani M, Aldosari Z, Aldosari O, Almhmd AE, Binsaleh S, Almannie R. A Narrative Review of the Toxic Effects on Male Reproductive and Sexual Health of Chewing the Psychostimulant, Catha edulis (Khat). Med Sci Monit. 2023 Apr 1;29:e939455. doi: 10.12659/MSM.939455. PMID: 37002591; PMCID: PMC10075001
Therefore, is this Khat consuming relevant for international readers, and for the article's subject with p = 0,015? Also, the Monthly income is statistically irrelevant for the main variables of the study.
In the table's titles, there is no explicit mention of the use of chi-square, and Cohen's kappa statistics. Mean, or standard deviation, is mentioned in the text, but not used.
In Table 2 in the Smoking paragraph, the second row - Previous smoking has a calculation mistake: 23+19=42, not 32
Related to the study limitation, a mention must be made: not only anthropometric data, but also all self-reported data have trust issues.
In the reference section, instead of print and electronic, it would be more useful in following up on the article to add the DOI number.
Author Response
Comment: Several ideas are reiterated throughout the text. For instance, the link between misperception of body weight and unhealthy behaviours is mentioned multiple times (lines 34, 50, 56, and 58), which can cause redundancy rather than reinforcing the argument and leading to duplication and fragmented narrative structure.
Response: The author of the manuscript appreciates the constructive comments of the reviewer which aided in enhancing the reporting quality of the manuscript. Several modifications were applied to the revised manuscript while ensuring reduction of redundancy. It is agreed that the impact of misperception of body weight and unhealthy behaviour is emphasized in this manuscript. Although the concept of misperception is repeated on several occasions in the manuscript, presenting a different view throughout the text was ensured such as impact of satisfaction of body weight on health behaviour, impact of social medial on perception and satisfaction, impact of normalization of abnormal body weight on health behaviour, and correlating the misperception to the stages of transtheoretical model. An effort was made to avoid duplication and support the indicated arguments with the relevant references.
Comment: It is interesting to understand why 32 was the age you chose to divide the study sample. A brief explanation is needed.
Response: The author of the manuscript appreciate the comment of the reviewer. The sample was divided based on the estimated mean age of the sample. This was performed to allow dichotomizing the sample and allowing performing the cross-tabulation displayed in table two. We also confirm that there was a typo concerning the mean age and the dichotomization is now made as either 31 or younger or older than 31.
This is now reflected in the data analysis section of the revised manuscript and highlighted with yellow as the following:
‘All the measured demographic variables were either binary or categorical except for age which was dichotomized based on the estimated sample mean as either 31 or younger or older than 31.’
We also confirm rectifying the typo concerning the age dichotomization in tables 1 and 2.
Comment: Is there any regional cultural reasoning for explaining women's misperception of overweight as being a normal silhouette?
Response: The author of the manuscript appreciates the comment of the reviewer. The current analysis indicated that females are more likely to misperceive their body weight in comparison to males. This is similar to other studies which assessed body weight misperception according to gender. The following is added and highlighted with yellow in the discussion section of the revised manuscript:
‘The frequency of body weight misperception was higher among females in the current study in comparison to male. This is similar to the findings of other studies which assessed perception of body weight according to gender. For example, in a Saudi study which involved a sample of 334 residents in Saudi Arabia, it was concluded that the prevalence of body weight misperception was higher among female [31], which is similar to the findings of the current study. Additionally, another Nigerian study which involved a sample of 567 adults, it was also reported that body weight misperception was higher among females in comparison to males [32]. Similarly, studies that assessed body weight satisfaction and perception among younger populations suggested that females are more likely to be less satisfied by their body weight [33], and to overestimate their body weight [34].’
Comment: The Khat chewing was a completely new concept and needed to read a supplementary reference: Alzahrani MA, Alsahli MA, Alarifi FF, Hakami BO, Alkeraithe FW, Alhuqbani M, Aldosari Z, Aldosari O, Almhmd AE, Binsaleh S, Almannie R. A Narrative Review of the Toxic Effects on Male Reproductive and Sexual Health of Chewing the Psychostimulant, Catha edulis (Khat). Med Sci Monit. 2023 Apr 1;29:e939455. doi: 10.12659/MSM.939455. PMID: 37002591; PMCID: PMC10075001
Response: Khat is a risky substance that has a direct effect on health and lifestyle of individuals in the studied population and similar international populations. A new paragraph is now added to the introduction section to define the concept to the readers, to explain its impact on health and lifestyle, and to illustrate where it is practiced internationally. The paragraph is added to the introduction of the revised manuscript, supported with relevant references and highlighted with yellow as the following:
‘Exposure to risky substances which can affect lifestyle varies according to populations. One of the studied substances that can impact lifestyle is Khat chewing which is mainly practiced in eastern countries in Africa [16], and southwestern regions of the Arabian peninsula [17, 18]. Additionally, Khat chewing is also practiced among migrants living in Australia [19], USA [20], and European countries [21-23]. Khat has a stimulant characteristics as it contains cathinone. Additionally, Khat chewing is linked to several health consequences including mental, oral and gastric, cancer, metabolic and nutritional conditions [24, 25].’
Comment: Therefore, is this Khat consuming relevant for international readers, and for the article's subject with p = 0,015? Also, the Monthly income is statistically irrelevant for the main variables of the study.
Response: The author of the manuscript agrees with the comment of the reviewer. As per the suggestions of the reviewer, Khat chewing is now clearly introduced to the reader and how it impact lifestyle as a risky substance. Additionally, the findings of the current investigation linking between khat chewing and body weight perception is added to the discussion section of the revised manuscript and highlighted with yellow as the following:
‘Studies that assessed body weight misperception according to khat chewing are lacking. However, khat chewing is a well-known risk factor for lower BMI among khat chewers [43, 44]. Similarly, Khat is known for its impact on appetite suppression [45], and the higher risk of under nutrition [44].‘
The lack of association of income with the study variables, in addition to other demographic variables is indicated in the results section. This is highlighted with yellow in the results section of revised manuscript as the following:
‘Other demographic factors such as age, education level, area of residence, and monthly income were not associated with the misperception of body weight in the current sample.’
Comment: In the table's titles, there is no explicit mention of the use of chi-square, and Cohen's kappa statistics. Mean, or standard deviation, is mentioned in the text, but not used.
Response: Use of chi-squared test is now declared in table 2 footer and highlighted with yellow in the revised manuscript. Kappa value is now declared in the table 3 footer and highlighted with yellow in the revised manuscript. Mean and standard deviation findings are presented in the first paragraph of the results section and highlighted with yellow in the revised manuscript as the following:
‘The mean age of the participants was 31.3 (SD: 11.1).’
Comment: In Table 2 in the Smoking paragraph, the second row - Previous smoking has a calculation mistake: 23+19=42, not 32
Response: The author of the manuscript appreciates the comment of the reviewer. The indicated typing mistake was identified and corrected in table two. The correction was highlighted with yellow in the revised manuscript. We also confirm that all tables have been checked for calculation accuracies and applied modifications were highlighted with yellow accordingly.
Comment: Related to the study limitation, a mention must be made: not only anthropometric data, but also all self-reported data have trust issues.
Response: The risk of reporting bias is acknowledged in the last paragraph of the discussion. The added text about self-reported data is added to the revised manuscript and highlighted with yellow as the following:
‘The main limitation relates to the inherent nature of self-reported weight and height which can be affected by higher risk of reporting bias in comparison to an objective clinical measurement’
Comment: In the reference section, instead of print and electronic, it would be more useful in following up on the article to add the DOI number.
Response: The author agrees with the comment of the reviewer concerning addition of the DOI numbers of the references. However, we confirm that the references were arranged via Endnote software according to the journal referencing guidelines and we agree to add the DOI numbers if approved by the editorial office.
Round 2
Reviewer 1 Report
Comments and Suggestions for Authors
The manuscript is much clearer to the reader.
Reviewer 2 Report
Comments and Suggestions for Authors
I recommend acceptance for publication in the present form.